# Learning from Crowds with Dual-View $K$-Nearest Neighbor

Jiao Li[1]  Liangxiao Jiang[*1]  Xue Wu[1]  Wenjun Zhang[1]

[1]School of Computer Science, China University of Geosciences, Wuhan 430074, China.

## Abstract

In crowdsourcing scenarios, we can obtain multiple noisy labels from different crowd workers for each instance and then infer its integrated label via label integration. To achieve better performance, some recently published label integration methods have attempted to exploit the multiple noisy labels of inferred instances' nearest neighbors via the $K$-nearest neighbor (KNN) algorithm. However, the used KNN algorithm searches inferred instances' nearest neighbors only relying on the defined distance functions in the original attribute view and totally ignoring the valuable information hidden in the multiple noisy labels, which limits their performance. Motivated by multi-view learning, we define the multiple noisy labels as another label view of instances and propose to search inferred instances' nearest neighbors using the joint information from both the original attribute view and the multiple noisy label view. To this end, we propose a novel label integration method called dual-view $K$-nearest neighbor (DVKNN). In DVKNN, we first define a new distance function to search the $K$-nearest neighbors of an inferred instance. Then, we define a fine-grained weight for each noisy label from each neighbor. Finally, we perform weighted majority voting (WMV) on all these noisy labels to obtain the integrated label of the inferred instance. Extensive experiments validate the effectiveness and rationality of DVKNN.

## 1 INTRODUCTION

Due to the capacity to learn from labeled data, supervised learning has been demonstrated its remarkable power in various fields [Jiang et al., 2019, Zhang et al., 2023a]. How-

ever, the success of supervised learning heavily relies on a substantial amount of high-quality labeled data for effective model training [Wang and Wu, 2021, Chen et al., 2022b, Hu et al., 2023]. Although the traditional approach to label collection by experts in the field or well-trained workers can receive the high reliability of the labels, it entails considerable investment in time and economy [Li et al., 2016, Xu et al., 2021, Zhu et al., 2023].

Fortunately, the advent of crowdsourcing platforms [Buhrmester et al., 2011] like Crowdflower, Clickworker, and Amazon Mechanical Turk has presented an economical and convenient alternative for collecting large-scale labels [Li et al., 2019, Dizaji et al., 2020, Zhang, 2022]. On these crowdsourcing platforms, labeling tasks can be published to a broad range of non-expert crowd workers at a low cost [Chen et al., 2022a, Tong et al., 2020]. However, because of the lack of professional experience and specific training, the labels provided by an individual worker are hard to meet the same quality standards as those from experts [Rodrigues and Pereira, 2018, Chen et al., 2020]. To mitigate this issue, *repeated labeling* [Sheng et al., 2008] has been proposed, which requires different crowd workers to provide multiple noisy labels for each instance and then infers its integrated label via label integration [Ma et al., 2015].

In the past few years, a large number of label integration methods have been proposed from various perspectives. Among these methods, majority voting (MV) is the simplest but often effective one. MV assumes that each crowd worker has the same labeling quality and regards the label with the highest votes as the integrated label. Obviously, the assumption of MV is not possible in real-world scenarios. That's why the performance of MV is suboptimal, and more sophisticated label integration methods have been proposed one after another. Representative works include: Dawid-Skene (DS) [Dawid and Skene, 1979], generative model of labels, abilities, and difficulties (GLAD) [Whitehill et al., 2009], ZenCrowd (ZC) [Demartini et al., 2012], Karger, Oh, and Shah (KOS) [Karger et al., 2014], iterative weighted majority voting (IWMV) [Li and Yu, 2014], ground truth

---
[*]Corresponding author, Liangxiao Jiang <ljiang@cug.edu.cn>

inference using clustering (GTIC) [Zhang et al., 2016], label similarity-based weighted soft majority voting (LSWSMV) [Tao et al., 2020], differential evolution-based weighted soft majority voting (DEWSMV) [Tao et al., 2021], attribute augmentation-based label integration (AALI) [Zhang et al., 2023b], instance redistribution-based label integration (IRLI) [Zhang et al., 2024], etc. Although these label integration methods are proposed from various perspectives, they usually achieve the inference only based on the limited information of the inferred instance itself. In recent years, some scholars have noticed this limitation and attempted to exploit the multiple noisy labels of inferred instances' nearest neighbors to achieve better performance. The representative works include: multiple noisy label distribution propagation (MNLDP) [Jiang et al., 2022] and label augmented and weighted majority voting (LAWMV) [Chen et al., 2022c], etc.

As far as we know, the $K$-nearest neighbor (KNN) algorithm [Cover and Hart, 1967] has been widely used to search the nearest neighbors by these label integration methods due to its popularity and robustness. However, the used KNN algorithm only considers the original attribute view and totally ignores the valuable information hidden in the multiple noisy labels, which limits the performance of these label integration methods. In recent years, multi-view learning [Sun, 2013, Zhao et al., 2017] develops rapidly and has achieved great success. Due to the complementary and consensus principle, multi-view learning is more effective than traditional single-view learning. Motivated by multi-view learning, we define the multiple noisy labels as another label view of instances and propose to search neighbors using the joint information from both the original attribute view and the multiple noisy label view.

To this end, we propose a novel label integration method called dual-view $K$-nearest neighbor (DVKNN). In DVKNN, we first define a new distance function to search the $K$-nearest neighbors of an inferred instance. Then, we define a fine-grained weight for each noisy label from each neighbor. Finally, we perform weighted majority voting (WMV) on all these noisy labels to obtain the integrated label of the inferred instance. In general, the contributions of this work can be summarized as follows:

1. We define the multiple noisy labels as another label view of instances and propose to search the nearest neighbors for an inferred instance using the joint information from both the original attribute view and the multiple noisy label view. On this basis, a new distance function is defined for crowdsourcing.

2. We propose a novel label integration method called dual-view $K$-nearest neighbor (DVKNN). In DVKNN, we use the new defined distance function to search the $K$-nearest neighbors for inferred instances and then define a fine-grained weight for each noisy label from

each neighbor, which enhances the effect of WMV and thus improves the performance of label integration.

3. We conduct extensive experiments to evaluate the proposed DVKNN on a large number of crowdsourced datasets. The experimental results show that DVKNN significantly outperforms all of its competitors.

The rest of this paper is organized as follows. Section 2 describes the related works. Section 3 introduces the proposed DVKNN in detail. Section 4 reports the experiments and results. Section 5 concludes this paper and outlines the research directions of future work.

## 2 RELATED WORK

In recent years, label integration has attracted a great deal of attention from scholars. A variety of label integration methods have been proposed from different perspectives [Sheng and Zhang, 2019]. MV [Sheng et al., 2008] is the simplest label integration method. In addition to MV, more complex label integration methods have been proposed one after another for better performance. For example, DS [Dawid and Skene, 1979] first introduces the EM algorithm [Singh, 2006] to deal with label integration, which uses the EM algorithm to jointly estimate the confusion matrix of each worker and the integrated label of each instance. KOS [Karger et al., 2014] estimates the integrated label of each instance and the labeling quality of each worker by belief propagation and low-rank matrix approximation. GTIC [Zhang et al., 2016] uses the K-means algorithm to infer the integrated labels, which first clusters all instances into distinct clusters and subsequently assigns the same class label to instances within the same cluster. DEWSMV [Tao et al., 2021] defines three objective functions to optimize the labeling qualities of workers by a differential evolution (DE) algorithm, which provides a novel perspective to handle label integration. AALI [Zhang et al., 2023b] designs an attribute augmentation method to enrich the original attribute space and builds multiple component classifiers on reliable instances to predict the integrated label for each instance.

In recent years, some scholars have noticed that existing label integration methods usually achieve the inference only based on the limited information of the inferred instance itself, so it is hard for them to achieve breakthrough performance improvements. Therefore, some recently published label integration methods have attempted to exploit the multiple noisy labels of inferred instances' nearest neighbors via the KNN algorithm. For example, Jiang et al. [2022] propose MNLDP. MNLDP first uses multiple noisy labels to calculate the multiple noisy label distribution for each instance. Then, MNLDP searches the nearest neighbors for the inferred instance in the attribute space and optimizes the weights of the nearest neighbors by the locally linear embedding. Finally, MNLDP uses label distribution propagation to

combine the multiple noisy label distributions of the inferred instance and its nearest neighbors to infer the integrated label. Chen et al. [2022c] propose LAWMV. LAWMV first searches neighbors in the attribute space to augment each instance's multiple noisy labels via the KNN algorithm. Then, LAWMV uses the distance and label similarity to weight the labels from different neighbors. Finally, LAWMV uses weighted majority voting to obtain the integrated label.

These label integration methods, which exploit the multiple noisy labels of inferred instances' nearest neighbors, have indeed improved the performance of label integration. However, the used KNN algorithm only considers the original attribute view and totally ignores the valuable information hidden in the multiple noisy labels, which limits their performance. Therefore, motivated by multi-view learning, we define the multiple noisy labels as another label view of instances and propose a novel label integration method called dual-view $K$-nearest neighbor (DVKNN). We will describe DVKNN in detail in Section 3.

## 3 DUAL-VIEW $K$-NEAREST NEIGHBOR

### 3.1 MOTIVATION

A crowdsourced dataset is typically denoted by $D = \{(\boldsymbol{x}_i, \boldsymbol{L}_i)\}_{i=1}^{N}$, where $N$ denotes the total number of instances. Each instance, denoted as $\boldsymbol{x}_i$, consists of $M$ attribute values $\{x_{im}\}_{m=1}^{M}$ and is associated with a multiple noisy label set $\boldsymbol{L}_i = \{l_{ir}\}_{r=1}^{R}$, where $R$ denotes the total number of crowd workers, and $l_{ir}$ denotes the label assigned to $\boldsymbol{x}_i$ by the $r$-th worker $u_r$. Each label takes the value from a fixed set $\{c_1, c_2, \cdots, c_Q, -1\}$, where $Q$ denotes the number of all classes and -1 denotes that $u_r$ does not annotate $\boldsymbol{x}_i$. To infer the unknown true label $y_i$ for $\boldsymbol{x}_i$, label integration is usually used to infer an integrated label $\hat{y}_i$, which is expected to be as consistent as possible with $y_i$.

To the best of our knowledge, to achieve better performance, some recently published label integration methods have attempted to exploit the multiple noisy labels of inferred instances' nearest neighbors via the KNN algorithm. However, as mentioned above, the used KNN algorithm only relies on the defined distance functions in the original attribute view and totally ignores the valuable information hidden in the multiple noisy labels. In crowdsourcing scenarios, multiple noisy labels contain a lot of valuable information, which can also describe the characteristics of instances. Therefore, totally ignoring the information from multiple noisy labels may have limited the performance of these label integration methods. So in this paper, we aim to make full use of the information from not only the original attribute view but also the multiple noisy labels to search the nearest neighbors for the inferred instance more accurately.

We have noticed that, in recent years, multi-view learning

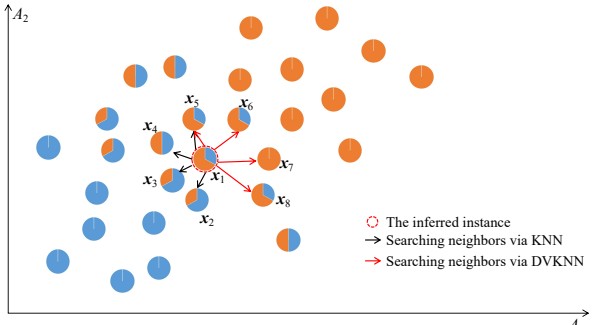

Figure 1: Difference between KNN and DVKNN.

develops rapidly and has achieved great success. Multi-view learning is an important learning framework in machine learning [Sun, 2013, Zhao et al., 2017]. It aims to model the information that can be obtained from multiple views. Due to the complementary and consensus principle, multi-view learning is more effective than traditional single-view learning. Here, the complementary principle emphasizes that the information provided by different views is complementary to each other. This means that each view may focus on a different aspect, which together constitutes a complete description of the dataset. And the consensus principle emphasizes that although the information from different views may differ, they should reflect the essential consistency of the same dataset at a high level. Motivated by multi-view learning, we define the multiple noisy labels as another label view of instances and then search inferred instances' nearest neighbors using the joint information from both the original attribute view and the multiple noisy label view. To this end, we propose a novel label integration method called dual-view $K$-nearest neighbor (DVKNN).

Figure 1 graphically shows the difference between KNN and DVKNN in searching neighbors, which roughly illustrates the effectiveness of DVKNN. In Figure 1, each circle represents an instance from a binary dataset with two attributes $A_1$ and $A_2$. The proportion of blue and orange in each circle represents the distribution of two classes, $c_1$ and $c_2$, in this instance's multiple noisy label set. We describe the difference between two instances in the original attribute view by the positional relationship on the axis, and in the multiple noisy label view by the proportion of color in the circle. We assume that the inferred instance is $\boldsymbol{x}_1$, and we search its five nearest neighbors (including itself) via KNN and DVKNN, respectively. As KNN only considers the original attribute view, the neighbors searched via KNN are $\boldsymbol{x}_1$, $\boldsymbol{x}_2$, $\boldsymbol{x}_3$, $\boldsymbol{x}_4$ and $\boldsymbol{x}_5$. In addition to the original attribute view, DVKNN also considers the multiple noisy label view, so the neighbors searched via DVKNN are $\boldsymbol{x}_1$, $\boldsymbol{x}_5$, $\boldsymbol{x}_6$, $\boldsymbol{x}_7$ and $\boldsymbol{x}_8$. We can see that the multiple noisy label distributions of the neighbors searched via KNN are confusing. On the contrary, the multiple noisy label distributions of the neighbors searched via DVKNN show high consensus. In this case, the neighbors searched via DVKNN are more reliable for

inferring the integrated label of $\boldsymbol{x}_1$.

## 3.2 THE PROPOSED DVKNN

Figure 2 graphically shows the overall framework of DVKNN. From it, we can see that DVKNN mainly includes two parts: distance measure and fine-grained weighting.

### 3.2.1 Distance Measure

Distance measure is the crucial step of DVKNN. Therefore, the core problem we have to solve first is how to measure the distance between instances from both the original attribute view and the multiple noisy label view.

Firstly, we focus on the original attribute view. How to measure the distance from the original attribute view has actually been widely studied, and many distance functions have been proposed in the past few years. Here, we take advantage of the widely used heterogeneous Euclidean-overlap metric (HEOM) [Wilson and Martinez, 1997] to measure the distance $d^1(\boldsymbol{x}_i, \boldsymbol{x}_j)$ between two instance $\boldsymbol{x}_i$ and $\boldsymbol{x}_j$ from the original attribute view, which can be calculated by:

$$d^1(\boldsymbol{x}_i, \boldsymbol{x}_j) = \sqrt{\sum_{m=1}^{M} d_m(\boldsymbol{x}_i, \boldsymbol{x}_j)^2}, \qquad (1)$$

where $d_m(\boldsymbol{x}_i, \boldsymbol{x}_j)$ denotes the distance between the $m$-th attribute values, which can be calculated as follows:

$$d_m(\boldsymbol{x}_i, \boldsymbol{x}_j) = \begin{cases} 1, & x_{im} \text{ or } x_{jm} \text{ is unknown} \\ overlap(x_{im}, x_{jm}), & x_{im} \text{ is nominal} \\ rn\_diff(x_{im}, x_{jm}), & \text{otherwise} \end{cases}, \qquad (2)$$

where $overlap(x_{im}, x_{jm})$ and $rn\_diff(x_{im}, x_{jm})$ are the differences between two nominal and numerical attribute values, respectively. They can be calculated by:

$$overlap(x_{im}, x_{jm}) = \begin{cases} 0, \text{ if } x_{im} = x_{jm} \\ 1, \text{ otherwise} \end{cases}, \qquad (3)$$

$$rn\_diff(x_{im}, x_{jm}) = \frac{|x_{im} - x_{jm}|}{max_m - min_m}, \qquad (4)$$

where $max_m$ and $min_m$ are the maximum and minimum values of the $m$-th attribute observed in data, respectively.

Then, let's focus on the multiple noisy label view. In crowdsourcing scenarios, since crowd workers do not provide labels to all instances, there are usually too many missing labels from crowd workers in the multiple noisy label sets.

This makes it challenging for us to measure the distance between instances in the multiple noisy label view accurately. To address this issue, we convert the multiple noisy labels of each instance into a multiple noisy label distribution. In this way, the nominal labels are converted into numerical values, and the problem of missing labels can be solved without loss of information. Specifically, we can calculate the multiple noisy label distribution $\boldsymbol{P}_i = \{p_{iq}\}_{q=1}^{Q}$ of $\boldsymbol{x}_i$ by estimating its class membership probabilities using its multiple noisy labels as follows:

$$p_{iq} = \frac{\sum_{r=1}^{R} I(l_{ir} = c_q)}{\sum_{r=1}^{R} I(l_{ir} \neq -1)}, \qquad (5)$$

where $I(\cdot)$ is an indicator function that outputs 1 when the condition in parentheses is true, and 0 otherwise. Now, we can use the distance functions defined for numerical values to measure the distance between the multiple noisy label distributions of each pair of instances. To be consistent with the distance function used in the original attribute view, we use Euclidean distance to measure the distance $d^2(\boldsymbol{x}_i, \boldsymbol{x}_j)$ between $\boldsymbol{x}_i$ and $\boldsymbol{x}_j$ from the multiple noisy label view, which can be calculated as follows:

$$d^2(\boldsymbol{x}_i, \boldsymbol{x}_j) = \sqrt{\sum_{q=1}^{Q} (p_{iq} - p_{jq})^2}. \qquad (6)$$

Now we can obtain the distances between two instances from the original attribute view and the multiple noisy label view, respectively. According to the complementary and consensus principle of multi-view learning, we believe that the information fused from two views is more useful than any single view. So we fuse these two distances to define the ultimate distance $d(\boldsymbol{x}_i, \boldsymbol{x}_j)$ between $\boldsymbol{x}_i$ and $\boldsymbol{x}_j$ as:

$$d(\boldsymbol{x}_i, \boldsymbol{x}_j) = \alpha \cdot d^1(\boldsymbol{x}_i, \boldsymbol{x}_j) + (1 - \alpha) \cdot d^2(\boldsymbol{x}_i, \boldsymbol{x}_j), \quad (7)$$

where $\alpha$ is a controlling factor that adjusts the proportion of $d^1(\boldsymbol{x}_i, \boldsymbol{x}_j)$ and $d^2(\boldsymbol{x}_i, \boldsymbol{x}_j)$. When $\alpha$ is greater than 0.5, the distance between instances is more inclined to be influenced by the original attribute view and otherwise inclined to the multiple noisy label view.

Based on the new defined distance function, we can search $\boldsymbol{x}_i$'s $K$-nearest neighbors $\{\boldsymbol{x}_{ik}\}_{k=1}^{K}$ (including itself) by sorting the distances $\{d(\boldsymbol{x}_i, \boldsymbol{x}_j)\}_{j=1}^{N}$. In this paper, we set the number of the nearest neighbors $K$ as $\beta \cdot \frac{N}{Q}$, where $\frac{N}{Q}$ roughly estimates the number of instances per class and $\beta$ is a predefined scaling factor.

### 3.2.2 Fine-grained Weighting

After searching $K$-nearest neighbors, how to make full use of the noisy labels from these neighbors is the next core problem we need to solve. On the one hand, the noisy labels from different neighbors should have different degrees

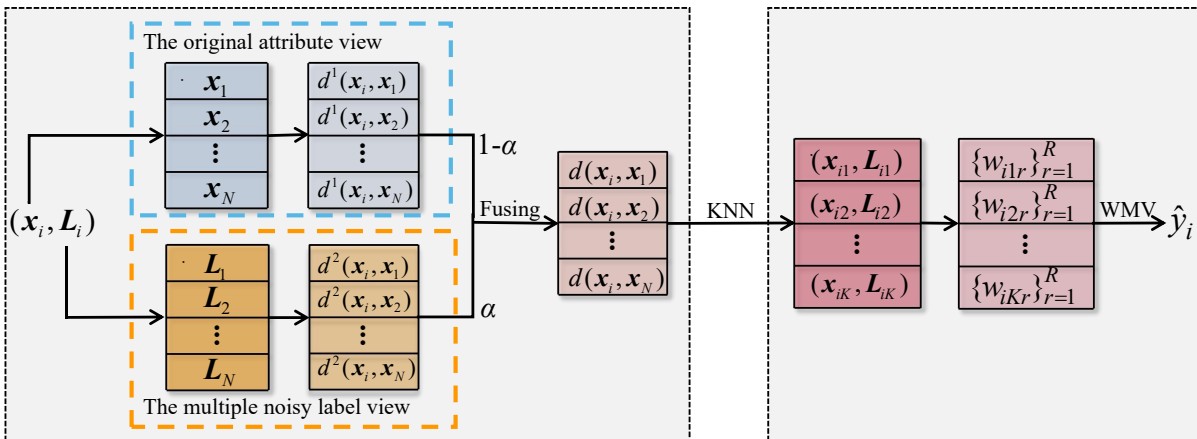

Figure 2: Overall framework of DVKNN.

of influence in inferring the integrated label. Specifically, the noisy labels from closer neighbors should have higher weights. On the other hand, the noisy labels from the same neighbor but different workers should also have different degrees of influence in inferring the integrated label. Specifically, the noisy labels from the workers with higher labeling qualities should have higher weights. Therefore, we define a fine-grained weight for each noisy label from each neighbor from the perspective of both instance and worker. For the label $l_{ikr}$ annotated by $u_r$ on the $k$-th neighbor of $\boldsymbol{x}_i$, its weight $w_{ikr}$ can be calculated as follows:

$$w_{ikr} = \left(1 - \frac{d(\boldsymbol{x}_i, \boldsymbol{x}_{ik})}{d(\boldsymbol{x}_i, \boldsymbol{x}_{iK})}\right) \cdot \frac{\sum_{r'=1}^{R} I(l_{ikr'} = l_{ikr})}{\sum_{r'=1}^{R} I(l_{ikr'} \neq -1)}, \quad (8)$$

where $d(\boldsymbol{x}_i, \boldsymbol{x}_{ik})$ is the distance between $\boldsymbol{x}_i$ and its $k$-th neighbor $\boldsymbol{x}_{ik}$ and $d(\boldsymbol{x}_i, \boldsymbol{x}_{iK})$ is the distance between $\boldsymbol{x}_i$ and its $K$-th neighbor $\boldsymbol{x}_{iK}$ with the maximum distance. $\sum_{r'=1}^{R} I(l_{ikr'} = l_{ikr})$ calculates the number of the labels on $\boldsymbol{x}_{ik}$ which are consistent with $l_{ikr}$ and $\sum_{r'=1}^{R} I(l_{ikr'} \neq -1)$ calculates the number of all labels on $\boldsymbol{x}_{ik}$.

From Equation (8), we can see that the former half, i.e. $1 - \frac{d(\boldsymbol{x}_i, \boldsymbol{x}_{ik})}{d(\boldsymbol{x}_i, \boldsymbol{x}_{iK})}$, is from the perspective of instance; the smaller the distance $d(\boldsymbol{x}_i, \boldsymbol{x}_{ik})$, the greater the weight $w_{ikr}$. And the latter half, i.e. $\frac{\sum_{r'=1}^{R} I(l_{ikr'} = l_{ikr})}{\sum_{r'=1}^{R} I(l_{ikr'} \neq -1)}$, is from the perspective of worker; the higher the label consistency between the worker $u_r$ and other workers, the higher the worker's labeling quality, and thus the greater the weight $w_{ikr}$.

After calculating a fine-grained weight for each noisy label from each neighbor, we finally perform WMV to infer the integrated label as follows:

$$\hat{y}_i = \underset{c_q \in \{c_1, c_2, \cdots, c_Q\}}{\arg\max} \sum_{k=1}^{K} \sum_{r=1}^{R} I(l_{ikr} = c_q) \cdot w_{ikr}. \quad (9)$$

---

**Algorithm 1** DVKNN

**Input:** $D = \{(\boldsymbol{x}_i, \boldsymbol{L}_i)\}_{i=1}^{N}$ -a crowdsourced dataset; $\alpha, \beta$ -the predefined parameters
**Output:** $\{\hat{y}_i\}_{i=1}^{N}$ -the integrated labels
1: **for** $i = 1$ to $N$ **do**
2:      **for** $j = 1$ to $N$ **do**
3:          Calculate the distance $d^1(\boldsymbol{x}_i, \boldsymbol{x}_j)$ from the original attribute view by Equations (1)-(4)
4:          Calculate the distance $d^2(\boldsymbol{x}_i, \boldsymbol{x}_j)$ from the multiple noisy label view by Equations (5)-(6)
5:          Calculate the distance $d(\boldsymbol{x}_i, \boldsymbol{x}_j)$ by Equation (7)
6:      **end for**
7:      Search $\boldsymbol{x}_i$'s $K$-nearest neighbors $\{\boldsymbol{x}_{ik}\}_{k=1}^{K}$ by sorting the distances $\{d(\boldsymbol{x}_i, \boldsymbol{x}_j)\}_{j=1}^{N}$
8:      **for** $k = 1$ to $K$ **do**
9:          **for** $r = 1$ to $R$ **do**
10:           Calculate $w_{ikr}$ by Equation (8)
11:          **end for**
12:      **end for**
13:      Infer the integrated label $\hat{y}_i$ of $\boldsymbol{x}_i$ by Equation (9)
14: **end for**
15: **return** $\{\hat{y}_i\}_{i=1}^{N}$

---

### 3.3 TIME COMPLEXITY ANALYSIS

Algorithm 1 describes the whole learning process of DVKNN. As shown in Algorithm 1, for each inferred instance, the time complexity of DVKNN can be divided into two parts: distance measure in lines 2-7, fine-grained weighting in lines 8-13.

In the first part, there includes: 1) Calculating the distances from the original attribute view; the time complexity is $O(NM)$. 2) Calculating the distances from the multiple noisy label view; the time complexity is $O(N(RQ + Q))$.

3) Fusing the distances from two views to define the ultimate distance; the time complexity is $O(N)$. 4) Searching neighbors; the time complexity is $O(NlogN)$. So the time complexity of the first part is $O(N(M+RQ+Q+1+logN))$.

In the second part, there includes : 1) Calculating a weight for each noisy label from each neighbor; the time complexity is $O(KR^2)$. 2) Performing weighted majority voting on all the noisy labels from $K$-nearest neighbors; the time complexity is $O(KR)$. So the time complexity of the second part is $O(KR^2 + KR)$.

To sum up, for each inferred instance, the time complexity of DVKNN is $O(N(M+RQ+Q+1+logN)+KR^2+KR)$. Therefore, the whole time complexity is $O(N^2(M+RQ+Q+1+logN)+N(KR^2+KR))$. If we only take the highest order terms, the time complexity of DVKNN is $O(N^2(M+RQ+Q+\log N))$.

# 4 EXPERIMENTS AND RESULTS

## 4.1 EXPERIMENTAL SETUP

In this section, we evaluate the performance of DVKNN on the crowd environment and its knowledge analysis (CEKA) [Zhang et al., 2015] platform. In our experiments, we use integration accuracy as the evaluation metric. Here, integration accuracy is the proportion of instances whose integrated labels are consistent with the ground true labels. We compare DVKNN with MV, GTIC, KOS, DEWSMV, AALI, MNLDP and LAWMV. We use the existing implementations of MV, GTIC and KOS on the CEKA platform and the implementations of DEWSMV, AALI, MNLDP and LAWMV provided by the original authors. All the parameter settings of these competitors are based on the corresponding papers. We also implement DVKNN on the CEKA platform and set the parameters $\alpha$ and $\beta$ to 0.7 and 0.6, respectively. To alleviate the fluctuation caused by the randomness, each experiment is repeated 10 times independently, and we report the averages as the final results.

## 4.2 EXPERIMENTS ON SIMULATED DATASETS

To evaluate the effectiveness of DVKNN, we conduct our experiments on the whole 34 simulated crowdsourced datasets published on the CEKA platform. Since some datasets have missing values and some competitors like MNLDP used in our experiments cannot handle missing values, we use the mean of numeric attribute values or the mode of nominal attribute values from the available data to replace all missing attribute values. After that, we simulate crowd workers to provide labels for instances. We first hide the ground true labels of all instances. Then, we simulate the labeling behavior of each crowd worker using two parameters: labeling ratio $p$ and labeling quality $q$, where $p$ denotes the probability that this worker annotates each instance, and $q$ denotes the probability that the labels provided by this worker are consistent with the ground true labels. In our simulated experiments, we set the number of crowd workers to 10. For each crowd worker, we randomly generate $p$ from a uniform distribution in the interval $[0, 1]$ and randomly generate $q$ from a uniform distribution in the interval $[0.55, 0.75]$.

Table 1 shows the detailed comparison results on the 34 simulated datasets. The highest accuracy corresponding to each dataset is highlighted in bold. The average accuracy of each method across 34 simulated datasets is also reported on the last row. Based on these results in Table 1, we then compare each pair of methods using the Wilcoxon signed-rank test [Demsar, 2006] and summarize the Wilcoxon test results in Table 2. In Table 2, the symbol ● indicates that the method in the row significantly outperforms the method in the corresponding column with the significance level 0.05, and the symbol ○ indicates that the method in the column significantly outperforms the method in the corresponding row with the significance level 0.1. These comparison results powerfully demonstrate the effectiveness of DVKNN. We can summarize the following highlights:

1. The average integration accuracies of MNLDP (89.47%), LAWMV (87.90%) and DVKNN (90.75%) on 34 datasets are much higher than those of MV (84.69%), GTIC (80.78%), KOS (82.91%), DEWSMV (84.29%), AALI (87.48%). These results prove that exploiting the multiple noisy labels of inferred instances' nearest neighbors is effective for improving the performance of label integration methods.

2. The average integration accuracy of DVKNN on 34 datasets is 90.75%, which is considerably higher than those of MNLDP (89.47%) and LAWMV (87.90%). These results prove that searching inferred instances' nearest neighbors using the joint information from both the original attribute view and the multiple noisy label view is more effective than only considering the original attribute view.

3. Based on the Wilcoxon test results, DVKNN significantly outperforms all of its competitors, which strongly validates the effectiveness of DVKNN.

Besides, we also conduct another group of experiments to verify the effectiveness of DVKNN for different labeling quality distributions, in which we randomly generate the labeling quality $q$ from a Gaussian distribution with $N(0.65, 0.1^2)$. The experiment results are shown in Tables 3 and 4. From these experimental results, we can draw to the conclusion that whether the simulated labeling quality of crowd worker belongs to a uniform or Gaussian distribution, DVKNN can notably outperform all the other label integration methods.

Table 1: Integration accuracy (%) comparisons on the uniform distribution.

| Dataset | MV | GTIC | KOS | DEWSMV | AALI | MNLDP | LAWMV | DVKNN |
|---|---|---|---|---|---|---|---|---|
| anneal | 92.83 | 62.45 | 94.25 | 92.97 | **94.34** | 94.21 | 84.52 | 89.19 |
| audiology | 77.57 | 75.88 | 69.12 | 76.81 | **80.44** | 75.58 | 76.64 | 79.91 |
| autos | 92.63 | 84.88 | 84.54 | 92.63 | 92.29 | 89.85 | 91.56 | **93.66** |
| balance-scale | 84.86 | 85.38 | 84.05 | 85.30 | 87.17 | **91.58** | 90.78 | 89.34 |
| biodeg | 81.37 | 83.23 | 82.49 | 81.46 | 85.95 | 87.18 | **87.78** | 87.65 |
| breast-cancer | 81.19 | 82.34 | 81.64 | 81.43 | **83.18** | 80.31 | 76.82 | 83.04 |
| breast-w | 83.22 | 84.45 | 84.35 | 83.05 | 92.60 | 95.34 | 95.54 | **95.99** |
| car | 87.63 | 86.28 | 89.06 | 87.27 | 88.47 | 88.58 | 80.88 | **90.05** |
| credit-a | 81.67 | 80.93 | 82.93 | 81.29 | 85.35 | 87.03 | 90.58 | **91.16** |
| credit-g | 81.13 | 82.95 | 82.13 | 80.76 | 83.48 | 79.96 | 76.60 | **85.31** |
| heart-c | 82.24 | 49.90 | 82.74 | 82.21 | 84.16 | 87.19 | 87.76 | **89.54** |
| heart-h | 84.18 | 48.47 | 84.86 | 84.12 | 84.80 | 88.20 | 88.06 | **90.78** |
| heart-statlog | 81.96 | 82.19 | 81.81 | 81.67 | 86.85 | 84.59 | 86.44 | **88.63** |
| hepatitis | 78.84 | 75.61 | 77.29 | 78.58 | 75.87 | 85.42 | 86.77 | **88.58** |
| horse-colic | 81.58 | 79.78 | 82.09 | 81.66 | 80.90 | 84.29 | 88.21 | **89.18** |
| hypothyroid | 89.93 | 63.39 | 93.45 | 89.74 | 90.51 | **95.56** | 92.29 | 92.29 |
| ionosphere | 81.51 | 79.89 | 81.77 | 80.83 | 84.84 | **89.23** | 78.03 | 87.29 |
| diabetes | 82.43 | 84.36 | 83.78 | 82.15 | 84.83 | 83.80 | 80.25 | **86.55** |
| iris | 90.40 | 90.93 | 84.60 | 90.87 | 96.07 | 98.47 | 98.40 | **99.00** |
| kr-vs-kp | 80.37 | 80.66 | 81.88 | 80.39 | 85.23 | 92.41 | 89.49 | **90.23** |
| labor | 76.14 | 73.68 | 55.09 | 76.14 | 82.11 | 86.32 | **88.60** | 88.07 |
| letter | 94.33 | 96.08 | 96.78 | 96.22 | 97.35 | **99.69** | 98.72 | 99.46 |
| lymph | 84.39 | 83.51 | 83.31 | 83.72 | 85.74 | 87.50 | 88.31 | **89.53** |
| mushroom | 83.14 | 83.39 | 84.37 | 83.22 | 85.88 | **98.95** | 91.70 | 98.19 |
| segment | 98.35 | 98.33 | 90.74 | 98.27 | 99.12 | 99.35 | 97.01 | **99.51** |
| sick | 79.37 | 71.83 | 81.24 | 79.40 | 86.26 | 92.83 | **93.88** | 93.85 |
| sonar | 79.13 | 79.90 | 79.04 | 78.80 | 77.93 | **86.39** | 80.48 | 84.95 |
| spambase | 80.49 | 82.14 | 82.12 | 80.60 | 87.21 | **89.37** | 84.63 | 85.19 |
| tic-tac-toe | 81.93 | 83.46 | 83.08 | 81.86 | 83.92 | 74.46 | 78.80 | **88.54** |
| vehicle | 93.45 | 93.46 | 90.05 | 93.42 | 94.33 | 93.18 | 90.41 | **95.96** |
| vote | 80.32 | 82.07 | 81.36 | 80.16 | 87.13 | 91.77 | 91.63 | **93.70** |
| vowel | 97.29 | 97.76 | 81.54 | 97.19 | 98.48 | **99.92** | 90.94 | 96.98 |
| waveform | 91.14 | 91.13 | 90.48 | 91.07 | 94.20 | 95.06 | **96.87** | 96.36 |
| zoo | 82.38 | 85.84 | 70.79 | 82.67 | 87.43 | 88.51 | **89.21** | 87.72 |
| **Average** | 84.69 | 80.78 | 82.91 | 84.65 | 87.48 | 89.47 | 87.90 | **90.75** |

Table 2: Integration accuracy (%) comparisons on the uniform distribution using the Wilcoxon test.

| | MV | GTIC | KOS | DEWSMV | AALI | MNLDP | LAWMV | DVKNN |
|---|---|---|---|---|---|---|---|---|
| MV | - | | | | ∘ | ∘ | ∘ | ∘ |
| GTIC | | - | | | ∘ | ∘ | ∘ | ∘ |
| KOS | | | - | | ∘ | ∘ | ∘ | ∘ |
| DEWSMV | | | | - | ∘ | ∘ | ∘ | ∘ |
| AALI | • | • | • | • | - | ∘ | | ∘ |
| MNLDP | • | • | • | • | • | - | | ∘ |
| LAWMV | • | • | • | • | | | - | ∘ |
| DVKNN | • | • | • | • | • | • | • | - |

Table 3: Integration accuracy (%) comparisons on the Gaussian distribution.

| Dataset | MV | GTIC | KOS | DEWSMV | AALI | MNLDP | LAWMV | DVKNN |
|---|---|---|---|---|---|---|---|---|
| anneal | 92.62 | 60.01 | 93.93 | 92.55 | **93.84** | **93.84** | 84.42 | 88.89 |
| audiology | 74.03 | 76.11 | 63.72 | 74.51 | **78.76** | 73.01 | 72.92 | 76.90 |
| autos | 91.90 | 86.10 | 82.44 | 92.00 | **93.02** | 88.78 | 91.27 | 92.34 |
| balance-scale | 87.42 | 87.49 | 86.06 | 86.98 | 90.05 | **91.90** | 90.51 | 89.68 |
| biodeg | 81.02 | 83.68 | 82.24 | 81.31 | 87.58 | 87.74 | **88.46** | 88.21 |
| breast-cancer | 83.71 | 84.76 | 84.62 | 83.60 | 84.83 | 83.04 | 78.29 | **85.38** |
| breast-w | 81.19 | 82.86 | 82.37 | 81.39 | 91.75 | 95.02 | 95.36 | **95.59** |
| car | 87.12 | 85.93 | 88.58 | 87.11 | 88.13 | 87.93 | 80.82 | **89.37** |
| credit-a | 79.91 | 78.67 | 80.99 | 79.68 | 83.09 | 85.28 | 89.84 | **90.33** |
| credit-g | 81.80 | 83.73 | 82.62 | 81.54 | 84.04 | 81.35 | 78.20 | **86.10** |
| heart-c | 82.44 | 48.84 | 82.38 | 81.82 | 84.26 | 86.47 | 87.33 | **89.04** |
| heart-h | 79.76 | 45.48 | 80.48 | 80.14 | 81.67 | 85.75 | 86.12 | **88.23** |
| heart-statlog | 77.44 | 78.37 | 70.74 | 77.07 | 83.15 | 83.37 | 85.26 | **87.07** |
| hepatitis | 78.65 | 73.87 | 73.94 | 78.00 | 73.16 | 85.81 | 86.52 | **88.00** |
| horse-colic | 81.17 | 79.35 | 81.77 | 81.09 | 79.21 | 84.32 | 86.85 | **88.70** |
| hypothyroid | 91.18 | 65.77 | 94.81 | 91.14 | 91.77 | **95.94** | 92.29 | 92.30 |
| ionosphere | 78.18 | 76.15 | 79.09 | 78.69 | 82.22 | 86.13 | 76.64 | **87.24** |
| diabetes | 82.25 | 84.05 | 83.80 | 82.20 | 84.60 | 83.67 | 79.74 | **86.39** |
| iris | 90.00 | 90.00 | 86.20 | 90.27 | 95.67 | 98.27 | **98.80** | 98.47 |
| kr-vs-kp | 80.29 | 80.71 | 81.58 | 80.42 | 86.12 | **92.14** | 88.86 | 89.74 |
| labor | 73.51 | 70.35 | 54.91 | 72.63 | 79.47 | 85.61 | 85.96 | **86.14** |
| letter | 94.23 | 96.23 | 96.99 | 97.02 | 97.56 | **99.60** | 98.67 | 99.54 |
| lymph | 80.41 | 80.34 | 79.19 | 80.88 | 83.04 | 86.76 | 85.68 | **87.23** |
| mushroom | 82.56 | 83.23 | 84.12 | 85.23 | 83.32 | **98.56** | 91.34 | 98.34 |
| segment | 96.66 | 96.60 | 87.57 | 96.65 | 97.82 | 99.10 | 97.02 | **99.30** |
| sick | 81.35 | 72.73 | 82.76 | 81.24 | 87.89 | **94.03** | 93.92 | 93.93 |
| sonar | 79.42 | 79.28 | 78.94 | 80.05 | 81.92 | **85.77** | 81.35 | 85.19 |
| spambase | 80.17 | 81.74 | 81.67 | 80.08 | 85.76 | **88.93** | 84.09 | 85.02 |
| tic-tac-toe | 79.06 | 80.75 | 79.70 | 78.53 | 81.54 | 74.59 | 78.33 | **85.94** |
| vehicle | 94.28 | 94.31 | 90.33 | 94.33 | 95.08 | 93.39 | 89.67 | **96.09** |
| vote | 82.87 | 84.48 | 83.66 | 82.99 | 87.49 | 92.99 | 92.23 | **94.16** |
| vowel | 96.20 | 96.93 | 81.90 | 96.18 | 97.83 | **99.71** | 86.38 | 95.17 |
| waveform | 91.40 | 91.35 | 90.95 | 91.18 | 94.14 | 95.07 | **97.02** | 96.44 |
| zoo | 80.50 | 86.04 | 68.22 | 80.40 | 85.54 | **88.61** | **88.61** | 87.33 |
| **Average** | 83.96 | 80.19 | 81.86 | 84.09 | 86.92 | 89.19 | 87.32 | **90.23** |

Table 4: Integration accuracy (%) comparisons on the Gaussian distribution using the Wilcoxon test.

| | MV | GTIC | KOS | DEWSMV | AALI | MNLDP | LAWMV | DVKNN |
|---|---|---|---|---|---|---|---|---|
| MV | - | | | | ○ | ○ | ○ | ○ |
| GTIC | | - | | | ○ | ○ | ○ | ○ |
| KOS | | | - | | ○ | ○ | ○ | ○ |
| DEWSMV | | | | - | ○ | ○ | ○ | ○ |
| AALI | • | • | • | • | - | ○ | | ○ |
| MNLDP | • | • | • | • | • | - | • | ○ |
| LAWMV | • | • | • | • | ○ | | - | ○ |
| DVKNN | • | • | • | • | • | • | • | - |

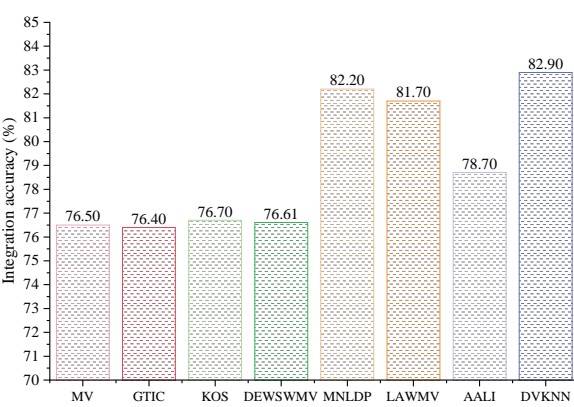

Figure 3: Integration accuracy (%) comparisons on the real-world dataset "LabelMe".

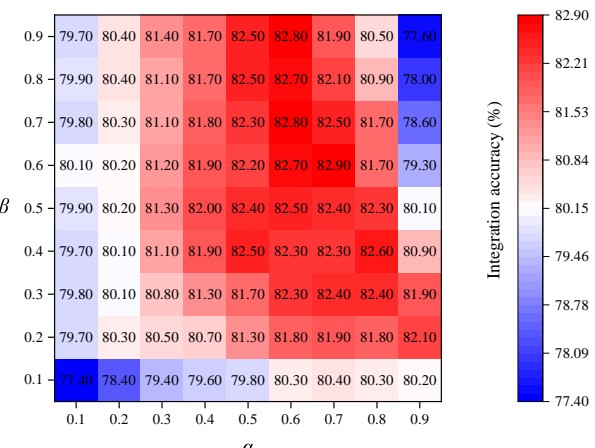

Figure 4: Integration accuracy (%) comparisons for DVKNN when $\alpha$ and $\beta$ vary from 0.1 to 0.9.

## 4.3 EXPERIMENTS ON REAL-WORLD DATASETS

To further evaluate the effectiveness of DVKNN, we also conduct our experiments on the real-world crowdsourced dataset "LabelMe" [Rodrigues et al., 2017], which is collected from Amazon Mechanical Turk (AMT) platform. The "LabelMe" dataset is a multi-class dataset used for classification, which contains 1000 instances described by 512 attributes. The AMT platform collects 2547 labels for this dataset from 59 different crowd workers. Figure 3 shows the detailed comparison results. The integration accuracy of DVKNN on the dataset "LabelMe" is 82.90%, which is significantly higher than that of MV (76.50%), GTIC (76.40%), KOS (76.70%), DEWSMV (76.61%), AALI (78.70%), MNLDP (82.20%) and LAWMV (82.90%). Based on these comparison results, we can draw the same conclusions as those on the simulated datasets, which further verifies the effectiveness of DVKNN.

## 4.4 PARAMETER SENSITIVITY ANALYSIS OF DVKNN

DVKNN has only two parameters: $\alpha$ and $\beta$. The parameter $\alpha$ adjusts the proportion of information from the original attribute view and the multiple noisy label view. As $\alpha$ increases, the influence of the original attribute view increases. The parameter $\beta$ adjusts the number of neighbors $K$. As $\beta$ increases, the number of neighbors $K$ increases. In this subsection, we conduct a group of experiments on the real-world dataset "LabelMe" to observe the integration accuracy of DVKNN when the parameters $\alpha$ and $\beta$ increase from 0.1 to 0.9. Figure 4 shows the detailed comparison results.

Firstly, we analyze the influence of the parameter $\alpha$ on the performance of DVKNN. From Figure 4, we can see that when $\alpha$ is higher than 0.5, DVKNN performs much

better than when $\alpha$ is lower than 0.5. When $\alpha$ is almost in the range $[0.6, 0.7]$, the integration accuracy of DVKNN reaches the highest. Then, when $\alpha$ increases from 0.7 to 0.9, the performance of DVKNN gradually deteriorates. These illustrate that the sightly more consideration on the original attribute view than the multiple noisy label view is more beneficial for DVKNN, and relying too much on the original attribute view is also detrimental to the performance of DVKNN. This is the reason why we set the parameter $\alpha$ to 0.7 in all of our previous experiments.

Then, we analyze the influence of the parameter $\beta$ on the performance of DVKNN. Theoretically, too small or too large $\beta$ are both detrimental to the performance of DVKNN. When $\beta$ is set too small, the information exploited from the neighbors is also too little. On the contrary, when $\beta$ is set too large, some irrelevant instances may be searched as neighbors. However, thanks to the fine-grained weighting for each noisy label from each neighbor, the negative impact carried by a large number of neighbors can be alleviated, so the performance degradation of DVKNN is not very obvious when the value of $\beta$ is large. As we can see in Figure 4, the performance of DVKNN is not very sensitive to the value of $\beta$ as long as it is not too small. Therefore, for simplicity, the value of $\beta$ can actually be directly set to 1 and thus be eliminated. In this case, the number of neighbors is directly set as $\frac{N}{Q}$. In this paper, we reserve the parameter $\beta$ and set its value to 0.6 in all of our previous experiments.

## 4.5 ABLATION EXPERIMENTS

To verify the effectiveness of each part of DVKNN, we compare DVKNN with its three variants in terms of integration accuracy. We denote its three variants DVKNN-a, DVKNN-l and DVKNN-w, which removes the original attribute view, the multiple noisy label view and the fine-grained weighting from DVKNN, respectively. Figure 5 shows the comparison results for DVKNN versus DVKNN-a, DVKNN-l and

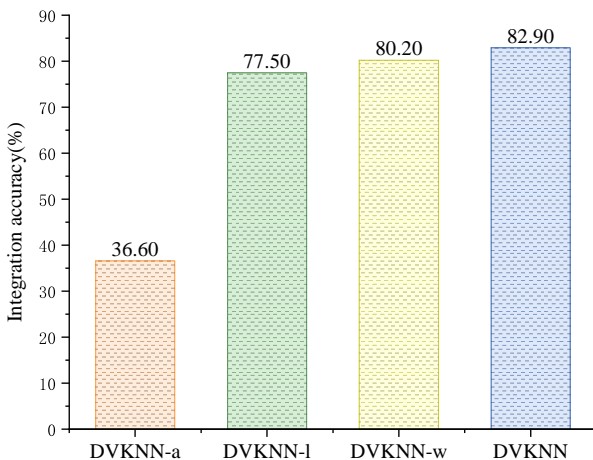

Figure 5: Integration accuracy (%) comparisons for DVKNN versus DVKNN-a, DVKNN-l and DVKNN-w.

DVKNN-w. As we can see, the integration accuracy of DVKNN (82.90%) is much higher than those of DVKNN-a (36.60%) and DVKNN-l (77.50%). Therefore, we can conclude that both the original attribute view and the multiple noisy label view are indispensable for DVKNN. In addition, the integration accuracy of DVKNN (82.90%) is also much higher than that of DVKNN-w (80.20%), which demonstrates that the fine-grained weighting for each noisy label from each neighbor is also effective for DVKNN. These results powerfully validate the effectiveness and rationality of each part of DVKNN.

## 5 CONCLUSIONS AND FUTURE WORK

In this paper, we find that the KNN algorithm used by some recently published label integration methods only consider the original attribute view and totally ignores the valuable information hidden in the multiple noisy labels, which limits the performance of these label integration methods. Motivated by multi-view learning, we define the multiple noisy labels as another label view of instances and propose a novel label integration method called dual-view $K$-nearest neighbor (DVKNN). In DVKNN, we define a new distance function using the joint information from both the original attribute view and the multiple noisy label view to search neighbors. Then, we define a fine-grained weight for each noisy label from each neighbor, which enhances the effectiveness of WMV and thus improve the performance of label integration. The extensive experimental results on a large number of crowdsourced datasets show that DVKNN significantly outperforms all of its state-of-the-art competitors.

In the current version of DVKNN, we artificially set the value of the controlling factor $\alpha$ based on the experimental experience, which is a little rough. Different datasets have different data characteristics, so the optimal value of $\alpha$ for different datasets may fluctuate within a suitable range. We believe that finding a better way to adaptively learn the value of $\alpha$ based on the data characteristics of the datasets can further improve DVKNN. This is an important research direction for our future work.

## Acknowledgements

We thank the anonymous reviewers for their helpful comments. The work was supported by National Natural Science Foundation of China (62276241).

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
