# OpenReview forum: "Learning from Crowds with Dual-View K-Nearest Neighbor"
_auai.org/UAI/2024/Conference — UAI 2024 poster_

### Official Review · Reviewer_wgWr · 2024-03-14

**Q2-1 Originality-Novelty:** 3
**Q2-2 Correctness-Technical Quality:** 3
**Q2-5 Clarity Of Writing:** 3

**Q1 Summary And Contributions:**

This paper finds a problem that the KNN algorithm used in many label integration methods ignores the information in the multiple noisy labels and thus may limit the performance of these label integration methods. To address this problem, this paper proposes a novel label integration method called dual-view K-nearest neighbor which treats the multiple noisy labels as a new view of instances. In this method, the authors define a new distance function based on the original feature and the multiple noisy labels, and then define a fine-grained weight for each noisy label from each neighbor. Finally, the authors conduct extensive experiments to validate the proposed methods.

**Q2-3 Extent To Which Claims Are Supported By Evidence:**

4: Excellent: all claims are supported by very convincing evidence (in the form of comprehensive experimental evaluation, rigorous mathematical proofs, detailed (pseudo-)code, precise references, well-motivated and realistic assumptions) and the authors deliver what they promise.

**Q2-4 Reproducibility:**

3: Good: key resources (e.g. proofs, code, data) are available and key details (e.g. proofs, experimental setup) are sufficiently well-described for competent researchers to confidently reproduce the main results.

**Q3 Main Strengths:**

First, in terms of motivation and methodology, this paper discusses the existing label integration methods in detail and points out that all existing KNN-based label integration methods only consider the original features and ignore the potential information in the label. Further, in order to address the above problem, this paper proposes a new label integration algorithm. This algorithm adopts a new distance function to obtain the nearest neighbors, and computes the fine-grained weights between the noisy labels and the neighbors, thus integrating the neighbors and the corresponding weights using the WMV method.
Second, in terms of experiments, this paper demonstrates the effectiveness of the proposed method through sufficient experiments. Finally, in terms of writing, this paper is easy to understand.

**Q4 Main Weakness:**

1）	There is a large blank area in the two gray boxes on the right side of Figure 2.
2）	The font size of Table 1 and Table 3 is too small.
3）	The references are not formatted consistently.

**Q5 Detailed Comments To The Authors:**

1）	Re-draw Figure 2 to make it look more compact.
2）	Increase the font size of Tables 1 and 3.
3）	Revise the formatting of the references to make them consistent.

**Q9 Complying With Reviewing Instructions:**

Yes

---

> ### Author Rebuttal · Authors · 2024-04-06
>
> Reviewer wgWr:
> 1)There is a large blank area in the two gray boxes on the right side of Figure 2.
> 2)The font size of Table 1 and Table 3 is too small.
> 3)The references are not formatted consistently.
>
> Author Response:
> Thanks for your valuable comments. In the final version, we will re-draw Figure 2 and increase the font size of Tables 1and 3 and format of the references consistently.

---

### Official Review · Reviewer_Vif5 · 2024-03-18

**Q2-1 Originality-Novelty:** 3
**Q2-2 Correctness-Technical Quality:** 4
**Q2-5 Clarity Of Writing:** 4

**Q1 Summary And Contributions:**

This paper proposes a new label integration method called DVKNN. Its main innovation is the introduction of the multi-view learning idea, which defines multiple noisy labels as multiple noisy label views of the instance and uses the joint information of the original attribute view and the multiple noisy label views to define a new distance function. In addition, DVKNN performs fine-grained weighting for each noisy label of each neighbor, improving the performance of label integration.

**Q2-3 Extent To Which Claims Are Supported By Evidence:**

3: Good: the main claims are supported by convincing evidence (in the form of adequate experimental evaluation, proofs, (pseudo-)code, references, assumptions).

**Q2-4 Reproducibility:**

4: Excellent: key resources (e.g. proofs, code, data) are available and key details (e.g. proof sketches, experimental setup) are comprehensively described for competent researchers to confidently and easily reproduce the main results.

**Q3 Main Strengths:**

1. The presentation of the paper is clear and the technical content is easy to follow. The ablation study is comprehensive.  It explains the enhancement effect of each module on the algorithm performance through experimental data.
2. Detailed analysis of the performance of DVKNN in all aspects, such as the time complexity of the algorithm and the accuracy of the algorithm integration.
3. Compared with previous label integration methods, the performance of the proposed method is significantly improved.

**Q4 Main Weakness:**

1. On the original features, the proposed method only uses one view. If multiple views are also introduced on the original features to describe the concept categories of samples from different perspectives, will the performance of label integration be further improved?
2. The author used two annotator quality distribution models, uniform distribution and Gaussian distribution, for simulation experiments. It would be better if the experiment could explore more annotator behavior patterns, such as the existence of spammer or adversary in the annotators.

**Q5 Detailed Comments To The Authors:**

This paper proposes a new label integration method called DVKNN. Its main innovation is the introduction of the multi-view learning idea, which defines multiple noisy labels as multiple noisy label views of the instance and uses the joint information of the original attribute view and the multiple noisy label views to define a new distance function. In addition, DVKNN performs fine-grained weighting for each noisy label of each neighbor, improving the performance of label integration. The experiment was conducted on 34 simulated crowdsourcing datasets and the LabelMe dataset on the CEKA platform, and the performance was significantly improved compared to existing label integration methods.

The most important contribution of the paper is demonstrating another approach that the joint utilization of original features and crowd labels can improve the label integration of crowdsourcing. The proposed Dual-View K-Nearest Neighbor model is novel and technically sound. The idea of the proposed method is clear and its analysis was well done. The authors conducted a set of experiments to show the effectiveness of the proposed method. Some other merits and weaknesses are listed in Q3 and Q4.

Overall, it is a nice piece of work. I recommend accepting the paper.

**Q9 Complying With Reviewing Instructions:**

Yes

---

> ### Author Rebuttal · Authors · 2024-04-06
>
> Reviewer Vif5:
> Q1: On the original features, the proposed method only uses one view. If multiple views are also introduced on the original features to describe the concept categories of samples from different perspectives, will the performance of label integration be further improved?
>
> Author Response:
> Thanks for your valuable comments. The fusion of the information from two views is a critical step in our proposed method. There are many fusion schemes can joint utilize the information of different views in the attributes. For example, one scheme is to augment the original attributes with multiple noisy labels and then use the augmented attributes to calculate the distances between instances directly, which is also available to merge the information from the original attribute view and the multiple noisy view. However, this scheme may lead to attribute redundancy. Specifically, the similar information in the original attributes and in the multiple noisy labels will be utilized redundantly, which leads to performance limitation. Thus, in DVKNN, we compute the distances in two views, respectively, and then use a hyperparameter alpha to fuse the distances in two views to define the distances between instances, which can avoid the information redundancy. Thanks again for your valuable comments.
>
> Reviewer Vif5:
> Q2: The author used two annotator quality distribution models, uniform distribution and Gaussian distribution, for simulation experiments. It would be better if the experiment could explore more annotator behavior patterns, such as the existence of spammer or adversary in the annotators.
>
> Author Response:
> Thanks for your valuable comments. Indeed, in the real-world crowdsourcing scenarios, the labeling patterns of annotators are complex and diverse. The existence of spammer or adversary in the annotators is possible. However, these special crowdsourcing scenarios with spammer and adversary are still challenging for general designed methods to achieve high performance without special considerations. Therefore, in this paper, we do not consider these special scenarios and only focus on the general crowdsourcing scenarios. To simulate the general real-world labeling patterns of annotators, we conduct two group of simulation experiments on different annotator quality distributions, i.e., uniform distribution and Gaussian distribution.
> Although we do not consider these special crowdsourcing scenarios in this version of paper, since them are interesting and valuable for research, we are pleased to discuss these special crowdsourcing scenarios and take them as our future work in the final version of this paper. Thanks again for your valuable comments.

---

### Official Review · Reviewer_5esP · 2024-03-23

**Q2-1 Originality-Novelty:** 3
**Q2-2 Correctness-Technical Quality:** 3
**Q2-5 Clarity Of Writing:** 3

**Q1 Summary And Contributions:**

This paper introduces a new label integration method, Dual-View K-Nearest Neighbor (DVKNN), which aims to improve the integration of multiple noisy labels in crowdsourcing by combining the original attribute view with the multiple noisy label view. The motivation is to address the limitation of existing KNN-based methods that rely solely on the original attribute view and ignore the information in noisy labels. This paper does extensive experiments on simulated and real-world datasets to show that DVKNN effectively enhances the accuracy of label integration.

**Q2-3 Extent To Which Claims Are Supported By Evidence:**

4: Excellent: all claims are supported by very convincing evidence (in the form of comprehensive experimental evaluation, rigorous mathematical proofs, detailed (pseudo-)code, precise references, well-motivated and realistic assumptions) and the authors deliver what they promise.

**Q2-4 Reproducibility:**

4: Excellent: key resources (e.g. proofs, code, data) are available and key details (e.g. proof sketches, experimental setup) are comprehensively described for competent researchers to confidently and easily reproduce the main results.

**Q3 Main Strengths:**

1. The idea of utilizing original attribute views and multiple noisy label views for neighbor searching, addressing limitations in existing KNN applications, is very novel.
2. The method has been validated across a broad range of crowdsourced datasets, including simulated and real-world datasets, ensuring its robustness and applicability.
3. The fine-grained weighting scheme is reasonable, further improving integration accuracy.

**Q4 Main Weakness:**

1. The requirement for a new distance measure, fine-grained weighting, and label integration could hinder its ease of use.
2. Due to its complex mechanism for weighing labels and integrating multiple noisy labels, there is a risk of overfitting on datasets with peculiar noise characteristics.
3. For large datasets with many classes or labels, the method could increase computational demands.

**Q5 Detailed Comments To The Authors:**

The work on DVKNN represents a significant step forward in leveraging crowdsourced data. There are a few questions:How does DVKNN perform in terms of computational efficiency and scalability, particularly with huge datasets or when handling a large number of classes? What are the computational limitations when dealing with big data?Could the authors explain the performance of DVKNN in scenarios with extremely noisy labels?

**Q9 Complying With Reviewing Instructions:**

Yes

---

> ### Author Rebuttal · Authors · 2024-04-06
>
> Reviewer 5esP:
> Q1: How does DVKNN perform in terms of computational efficiency and scalability, particularly with huge datasets or when handling a large number of classes? What are the computational limitations when dealing with big data?
>
> Author Response:
> Thanks for your valuable comments. Essentially, our proposed DVKNN is an improved KNN algorithm for label integration. It’s well known that KNN is a lazy learning method, which calculates the distance between each target instance and all instances in the whole dataset, so it has a high time complexity but good performance. Due to the characteristics of KNN, our DVKNN has also a relatively high time complexity when it handles huge datasets. We have provided the time complexity of the proposed DVKNN in subsection 3.3. For the part of our core idea in DVKNN, i.e., jointly utilizing the information of the original attribute view and the multiple noisy label view to define the distance between instances, it has actually little extra computational demand compared with the traditional KNN. However, because the two views are independent of each other, the distances in two views can be computed in parallel.
> In summary, compared with the traditional KNN for label integration, our DVKNN brings little extra time complexity, but significantly improves the effect of label integration. Thanks again for your valuable comments.
>
> Reviewer 5esP:
> Q2: Could the authors explain the performance of DVKNN in scenarios with extremely noisy labels?
>
> Author Response:
> Thanks for your valuable comments. For the performance of DVKNN in scenarios with extremely noisy labels, we believe it will be negatively affected by the extremely noisy labels mainly from the following aspects. (1) The reliability of the information from the multiple noisy label view decreases. In this case, we can deal with it by increasing the value of hyperparameter alpha to increase the influence of the original attribute view and reduce the influence of the multiple noisy label view. (2) The label consistency with other workers can hardly reflect the worker quality, so the reliability of the worker quality estimation decreases. We can deal with it by increasing the value of hyperparameter beta to increase the number of neighbors and then absorb more labels from them to reduce the negative influence.

---

### Official Review · Reviewer_CSAi · 2024-03-23

**Q2-1 Originality-Novelty:** 3
**Q2-2 Correctness-Technical Quality:** 3
**Q2-5 Clarity Of Writing:** 4

**Q10 Ethical Concerns:**

No.

**Q1 Summary And Contributions:**

The authors found that existing KNN-based crowdsourcing generally totally relies on the distance of feature space and totally ignores the noisy label information. To address that, they inspired by the multi-view learning and designed a novel label integration method called dual-view K-nearest neighbor (DVKNN). DVKNN integrates both the views of features and noisy labels. As reported in the experiments, DVKNN achieves satisfying performance.

**Q2-3 Extent To Which Claims Are Supported By Evidence:**

3: Good: the main claims are supported by convincing evidence (in the form of adequate experimental evaluation, proofs, (pseudo-)code, references, assumptions).

**Q2-4 Reproducibility:**

4: Excellent: key resources (e.g. proofs, code, data) are available and key details (e.g. proof sketches, experimental setup) are comprehensively described for competent researchers to confidently and easily reproduce the main results.

**Q3 Main Strengths:**

+ I believe this paper is well-written and easy to follow. The motivation of this paper is straightforward, that is to take into account the information hidden in the noisy labels for KNN when applied to crowdsourcing.

+ The DVKNN is novel by fusing the information from both views of features and noisy labels. The method is simple, yet as validated in the experiments, very effective to integrating the noisy label information.

+ The experiments are sufficient and persuasive. The performance results, such as Tables 1 and 3 well prove the better performance of DVKNN. On the other hand, the authors also conduct extensive statistical tests, such as Tables 2 and 4, well justify the significant better performance of DVKNN.

**Q4 Main Weakness:**

My concerns and questions are as follows:

- The distance fusion method, i.e., Eq. (7), weights two distance with a fixed $\alpha$. I wonder if it is possible to learn a different alpha for different x. The authors can give some discussion of that.

- How to set the number of nearest neighbors K?

**Q5 Detailed Comments To The Authors:**

Please refer to the Strengths and Weakness.

**Q9 Complying With Reviewing Instructions:**

Yes

---

> ### Author Rebuttal · Authors · 2024-04-06
>
> Reviewer CSAi:
> Q1: The distance fusion method, i.e., Eq. (7), weights two distance with a fixed alpha. I wonder if it is possible to learn a different alpha for different x. The authors can give some discussion of that.
>
> Author Response:
> Thanks for your valuable comment. The setting of the value of alpha is indeed a matter of concern in DVKNN. In the current version, we artificially set the value of alpha based on the experimental experience. In the Section 5 CONCLUSIONS AND FUTURE WORK, we have discussed that different datasets have different data characteristics, so the optimal value of alpha for different datasets may fluctuate within a suitable range. We believe that learn a different alpha for a different dataset could further improve our proposed method. However, the optimal value of alpha is affected by many factors, such as the attribute quality and the noise level of the labels, etc., which are both hard to be estimated in the real-world crowdsourcing scenarios where the ground true labels are unknown. Therefore, adaptively learning the value of alpha is still a challenging task, which is an important research direction for us in the future.
> In the final version, we will enhance the discussion about the setting of alpha in the part of future work to enrich our paper. Thanks again for your valuable comments.
>
> Reviewer CSAi:
> Q2: How to set the number of nearest neighbors K?
>
> Author Response:
> Thanks for your valuable comments. The setting of the number of nearest neighbors K is also a matter of concern in DVKNN. How to set the number of neighbors in KNN for label integration has been studied by some researches, such as [1], [2].
> [1] Label Augmented and Weighted Majority Voting for Crowdsourcing. Information Sciences, 2022, 606: 397-409.
> [2] FNNWV: Farthest-Nearest Neighbor-based Weighted Voting for Class-Imbalanced Crowdsourcing. Science China Information Sciences, doi: 10.1007/s11432-023-3854-7.
> In this paper, we have followed the solution of these researches. Specifically, we set the value of K by the equation $\beta \frac{N}{Q}$, where N/Q roughly estimates the number of instances of each class and \bata is a hyperparameter used to scale N/Q. As analysed in [1],  a larger value of K in the suitable range can reduce the influence of the noise labels. At the same time, K should not exceed the number of instances of each class. Since the ground true labels are unknown in the crowdsourcing scenarios, the equation mentioned above are adopted to adaptively set the number of neighbors instead of using a fixed value.
> In the final version, we will enhance the explanation of the setting of the number of nearest neighbors K in section 3.2.1. Thanks for your valuable comments again.

---

### Official Review · Reviewer_wSQN · 2024-03-23

**Q2-1 Originality-Novelty:** 3
**Q2-2 Correctness-Technical Quality:** 3
**Q2-5 Clarity Of Writing:** 4

**Q1 Summary And Contributions:**

This manuscript proposes duai-view knn method for addressing label integration called DVKNN. Specifically, DVKNN regards the multiple noisy labels as another label view of instances and searches inferred instances’ nearest neighbors using the joint information from both the original attribute view and the multiple noisy label view.

**Q2-3 Extent To Which Claims Are Supported By Evidence:**

3: Good: the main claims are supported by convincing evidence (in the form of adequate experimental evaluation, proofs, (pseudo-)code, references, assumptions).

**Q2-4 Reproducibility:**

4: Excellent: key resources (e.g. proofs, code, data) are available and key details (e.g. proof sketches, experimental setup) are comprehensively described for competent researchers to confidently and easily reproduce the main results.

**Q3 Main Strengths:**

1)	This manuscript regards the multiple noisy labels as another view and searches the nearest
neighbors jointly using the original attribute information and multiple noisy labels.
2)	A new distance is defined with fusing the attribute information and multiple noisy labels to search KNN for each instance.

**Q4 Main Weakness:**

My main concern is the fusion scheme of the attribute information and noisy label. It is known that the information provided by the noisy multiple noisy labels is typically not convincing. Is it appropriate to treat the \alpha according as the hyper-parameters and fuse these two-view information in average-weighted scheme?

**Q5 Detailed Comments To The Authors:**

Please see Q4.

**Q9 Complying With Reviewing Instructions:**

Yes

---

> ### Author Rebuttal · Authors · 2024-04-06
>
> Reviewer wSQN:
> My main concern is the fusion scheme of the attribute information and noisy label. It is known that the information provided by the noisy multiple noisy labels is typically not convincing. Is it appropriate to treat the \alpha according as the hyper-parameters and fuse these two-view information in average-weighted scheme?
>
> Author Response:
> Thanks for your valuable comments. Fusing information from the original attribute view and the multiple noisy label view is a key step in DVKNN. Indeed, due to the presence of noise in the multiple noisy labels, the information from the multiple noisy label view may be less reliable than the original attribute view. We have noticed that, although there are many schemes able to fuse the information of the two views, many of them treat the two views equally, which is unreasonable for DVKNN. Therefore, we adopt a simple but effective fusion scheme, which can use the hyperparameter \alpha to control the different influence of the two views. By increasing the value of \alpha, we can easily increase the influence of the original attribute view and decrease the influence of the multiple noisy label view. Actually, this fusion scheme has been used in many researches and achieved great performance, such as:
> [1] Learning from Crowds with Multiple Noisy Label Distribution Propagation. IEEE Transactions on Neural Networks and Learning Systems, 2022, 33(11): 6558-6568.
> [2] Dual-View Noise Correction for Crowdsourcing. IEEE Internet of Things Journal, 2023, 10(13): 11804-11812.
> All in all, although there are many schemes able to fuse the information of the two views, using hyperparameter \alpha to fuse information from two different views is a simple but effective scheme, which can control the different influence of two views conveniently and has been applied in many researches. That’s why we adopt it in the proposed method. Thanks again for your valuable comments.

---

### Meta-Review · Area_Chair_kHUA · 2024-04-18

All reviewers are unanimously in favor of accepting.

Additional comments: it seems that all p-values that are <0.1 are also <0.05, which to me sounds more like a consequence of using simulated data where the sample size can be arbitrarily large than anything to do with the practical significance of the differences between the methods. I encourage the authors to revisit the rationale of doing statistical tests in such a case.

Also, the 90s style patterning in the bar charts (Figs. 3 and 5) carries no information since the labels are indicated below each bar, and only serves to distract the reader; in line with principles of data visualization, the patterns should therefore be removed.